# Seasonal Variations in the Fish Species Composition and Community Structures on the Eastern Coast of South Korea

## Jeonghoon Han and Young-Ung Choi *

Marine Bio-Resources Research Unit, Korea Institute of Ocean Science & Technology (KIOST), Busan 49111, Korea
* Correspondence: yuchoi@kiost.ac.kr

**Abstract:** The seasonal variations in the species composition of set net fisheries were investigated on the Eastern coast of Korea, from 2007 to 2008. In total, 51 species were found that were classified into 15 orders and 33 families. The water temperature of the study area was 0.1–1.8 °C in 2007 and 0.6–2.1 °C in 2008, which was higher than the average water temperature in the previous decade. The monthly variation in the number of species peaked twice, in May (spring) and November (autumn) when the water temperature increased and decreased, respectively, and the monthly variation in the number of individuals showed a remarkably high trend in winter and autumn and was mainly caused by large migratory species. Based on the cluster analysis of the 18 most dominant species with more than 0.4% of the total number of individuals, we divided the species composition and community structures into three groups: fishes with a temporary appearance (Group A), fishes with a long-term appearance (Group B), and dominant pelagic fishes appearing with a long-term appearance (Group C). We could conclude that the seasonal variation in the structure of the fish community was mainly caused by the pelagic migration of species under high water temperature conditions during the study period.

**Keywords:** water temperature; fishery production; cluster analysis; dominant species; sustainable management

## 1. Introduction

The fish fauna in the East Sea of Korea was reported to consist of approximately 450 coastal settlement species, bathypelagic species, and migratory species. Except for the bathypelagic species, most fish species travel between the coast and the mainland and migrate seasonally from the south to the north regions [1–3]. These species include commercially important fish species (approximately 50 species), such as *Gadus macrocephalus*, *Theragra chalcogramma*, and *Clupea pallasii*, that are known as cold water fish, and cephalopods, such as *T. pacificus*, all of which are widely distributed on the Eastern coast of Korea [3]. However, the distribution of these representative species and the subsequent fishery production has changed due to climate change. The fishery production of *T. chalcogramma* comprised approximately 22% of the total production in the 1990s, which rapidly decreased to less than 1% in 2000. However, *T. pacificus* accounted for 26% of the total in 1990, which increased to 39% in 2000 [4], thus indicating that the fishery production of *T. pacificus* increased, but that of *T. chalcogramma* and *C. saira* decreased after the 1990s [3]. Accordingly, the fishery production on the Eastern coast of Korea increased from 150,000 metric tons (MT) in the early 1960s to 275,000 MT in the early 1980s; however, it decreased to 150,000 MT in the late 1980s, whereas in the 1990s, it frequently varied from 200,000 to 250,000 MT [5].

Previous studies have reported that large variations in the fishery biomass and production were caused by the short- and long-term influences of the climatic regime shifts, such as thermal front movements towards the north, the increase in the average surface ocean temperature, and the extended movement of the Kuroshio [6–8]. The Kuroshio warm

current intensifies the Tsushima warm current connected with the Kuroshio current in the inshore waters of Southern Korea, the increase in salinity of the South Sea of Korea from the Kuroshio current is presumed to have caused the increases in the recruitment and biomass of *Trachurus japonicus* with one of a major migratory fish species in the early 1980s [7,9]. In the 2000s, the scientific and industrial research on the Eastern coast of Korea had shifted focus to fishing grounds due to the irregular appearance of large migratory fishes and subtropical fishes that were difficult to observe in the previous decades [10]. The appearance of warm water fish species and subtropical fish species such as *Scomber japonicas*, *Engraulis japonicus*, and *T. pacificus* have been recently increasing [11,12]; therefore, multidimensional observations are necessary in order to assess these changes in the fish communities that occur due to climate regime shifts in the fishery grounds.

Gangwon-do is located on the central Eastern coast of Korea, where the sea conditions can change according to the changes in the location of the polar front [13]. The East Sea of Korea, including the coastal waters of Gangwon-do, has a unique hydrology, with a combination of both North Korean cold currents and Tsushima warm currents; therefore, the temporal and spatial distributions of the hydrological factors, such as water temperature and salinity, are complex [14]. The short- and long-term information on changes in the fishery catches in this region are required for understanding the characteristics of the fluctuations in fish resources. However, the information is limited for understanding the relationship between the seasonal variations in the environmental factors and the characteristics of fish catch variations in the central Eastern coast of Korea.

This study aimed to determine the seasonal variations of fish communities captured in set nets installed in the coastal waters of Yangyang, Gangwon-do, for two years in order to provide fundamental information on the temporal patterns of the fish species composition on the Eastern coast of the Korean Peninsula.

## 2. Materials and Methods

This study was conducted using three sets of set nets that were installed in order to capture fish from the middle to upper regions of the sea in Yangyang, on the east coast of Korea. The point of installation of these set nets was about 3.8 km from the coastline, and the depth of the water was about 50 m (37°58′27″ N, 128°47′18″ E; 37°58′21″ N, 128°47′24″ E; 37°58′18″ N, 128° 47′24″ E, respectively) (Figure 1). The dimensions of the fishing nets were 148 × 34.5 m (L × W), the length of the wing net was 64.4 mm, the mesh size was 75 × 75 mm, the length of the bag net was 35 m, and the mesh size was 33 mm × 33 mm (Figure 2). The fishing data from February 2007 to December 2008 were collected from logbooks that had compiled the information on set net fisheries at these three points. The daily sampling data recorded the number of species, the individuals, and the species diversity. Subsequently, a species checklist was created based on the number of species and individuals from each species. To identify fish species and their ecological characteristics, we used the studies by Nelson (1994) and Kim et al. (2005) as references [15,16], while the characteristics of the cephalopod distribution and identification were assessed using KORDI (2004) as a reference [17]. The daily sea surface temperature data, during the sampling period, was taken from the Sokcho Tide Station of the National Oceanographic Research Institute [18]. The daily number of individuals of the captured fishes was averaged for each month and subsequently, the monthly averages were compared in order to assess the quantitative changes. Further, the Shannon–Weiner index (H′; 1963) [19] was used in order to calculate the diversity of the fish species with the software PRIMER 6 (Primer-E Ltd., Plymouth, UK).

Out of the 51 species recorded, the data from the 18 most abundant species, that is, the species with more than 0.4% of the total abundance, were used for cluster analysis based on the temporal distribution of the fish species composition and their abundance. The cluster analysis was performed in order to classify the homogeneous groups within the sampling time points, in order to establish groupings and to recognize the relationship between them. The Q-mode (sampling time point) and R-mode (species) dissimilarity matrices

were computed on the square root transformed species dataset, using the MVSP shareware computer package (Multivariate Statistical Package, MVSP Shareware 3.0, Pentraeth, Isle of Anglesey, Wales, UK).

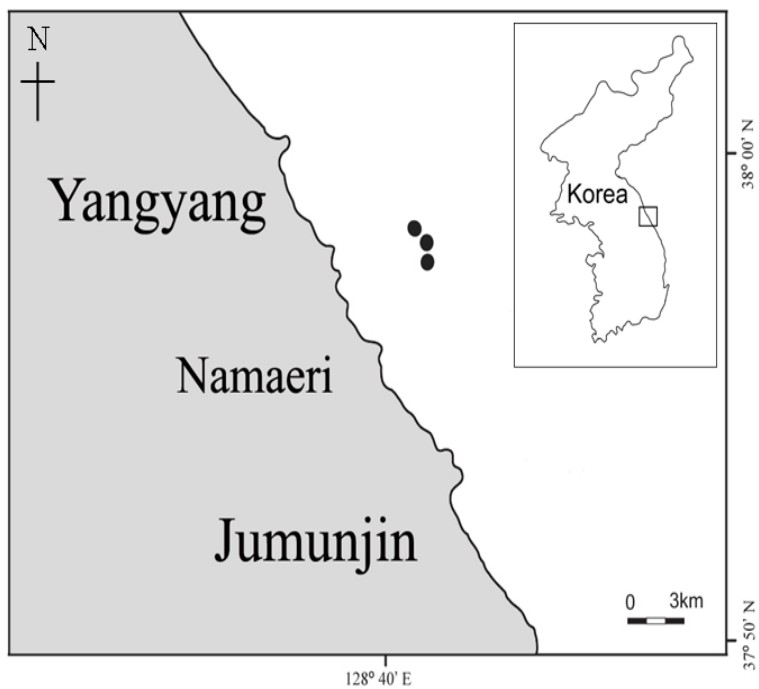

**Figure 1.** Map showing the location of the set nets in the coastal waters of Yangyang.

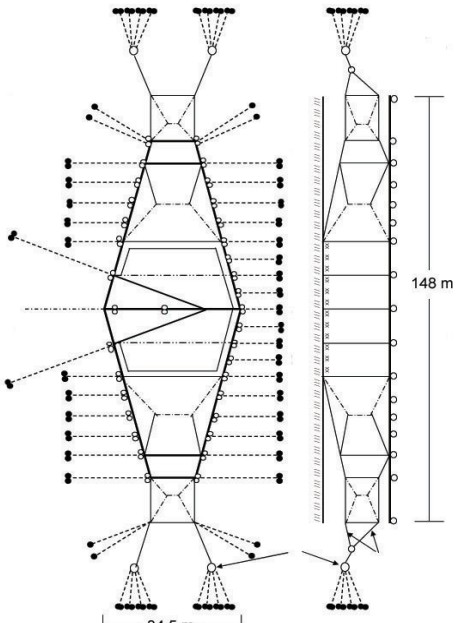

**Figure 2.** Schematic diagram of the set net fishery structure.

## 3. Results

### 3.1. Water Temperature

The daily average water temperature fluctuated from 7.8 °C to 22.9 °C in 2007 and from 8.0 °C to 23.8 °C in 2008. The temperature was at its lowest from January to February (7.8–8.2 °C). Subsequently, the temperature gradually increased during March–August, when the water temperature was recorded at 20 °C and above. The highest water temperature was recorded at 22.9 °C in August 2007 and at 23.8 °C in August 2008. The temperature

decreased from October onwards in both years, reaching 10 °C in December 2008. The water temperature was overall higher by 0.1–1.8 °C in 2007 and by 0.6–2.1 °C in 2008 (except for 0.3 °C lower in July and 0.7 °C lower in November) than the average water temperature in the previous decade (Figure 3).

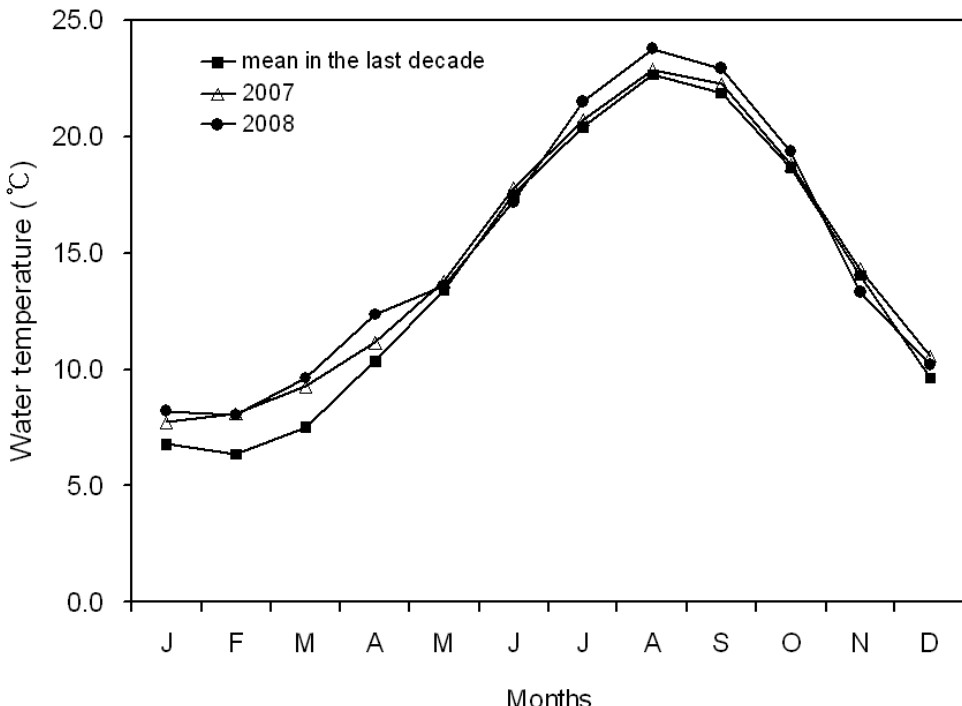

**Figure 3.** Monthly variations in the mean water temperature during the study period (2007–2008) in the coastal waters of Yangyang and between 1999 and2008 in the coastal waters of Sokcho.

*3.2. Species Composition*

During the study period, 51 species belonging to 35 families in 17 orders were observed in the set nets in the study area. A total of 48 fish species from 33 families in 15 orders and three cephalopod species belonging to two families in two orders were recorded in 2007 and 2008 (Table 1 and Supplementary Table S1). In total, 1,402,709 individual fish were captured during the entire study period.

**Table 1.** Fish and cephalopod species captured in set nets in the coastal waters of Yangyang from February 2007 to December 2008.

|  | Order | Family | Genus | Species |
|---|---|---|---|---|
|  | *Carcharhiniformes* | 1 | 1 | 1 |
|  | *Anguilliformes* | 1 | 1 | 1 |
|  | *Clupeiformes* | 2 | 3 | 3 |
|  | *Cypriniformes* | 1 | 1 | 1 |
|  | *Sakmoniformes* | 1 | 1 | 2 |
|  | *Gardiformes* | 1 | 1 | 1 |
|  | *Lophiiformes* | 1 | 1 | 1 |
| **Fishes** | *Atheriniformes* | 1 | 1 | 1 |
|  | *Beloniformes* | 1 | 2 | 2 |
|  | *Beryciformes* | 1 | 1 | 1 |
|  | *Gasterosteiformes* | 1 | 1 | 1 |
|  | *Scorpaeniformes* | 4 | 5 | 5 |
|  | *Perciformes* | 12 | 20 | 21 |
|  | *Pleuronectiformes* | 2 | 3 | 3 |
|  | *Tetraodoniformes* | 3 | 4 | 4 |

**Table 1.** *Cont.*

|  | Order | Family | Genus | Species |
|---|---|---|---|---|
| **Cephalopods** | *Octopoda* | 1 | 1 | 1 |
|  | *Teuthoidea* | 1 | 2 | 2 |
| | Total | 35 | 49 | 51 |

Fish and cephalopods accounted for 49.6% and 50.4% of the total species, respectively (Supplementary Table S1). The order Perciformes had the highest appearance rate (21 species from 12 families), followed by the order Scorpaeniformes (five species from four families), the order Tetraodontiformes (four species from three families), the order Clupeiformes (three species from two families), and the order Pleuronectiformes (three species from two families) (Table 1).

*3.3. Monthly Variations in Fish Species*

The number of daily recorded species per month was 3–11 species (Figure 4). A high number of species was observed from April to June in both years, with 10–11 species in 2007 and, 9–11 species in 2008. Further, the number of species peaked in May in both years. Another peak was recorded in November, with nine species and 13 species in 2007 and 2008, respectively. Moreover, a low number of species was recorded in February (four) and July (five) 2007 and in February (five) and August (four) 2008.

In 2007, the average daily number ranged from 518 individuals in July to 10,562 individuals in February (Figure 4). February, March, and October recorded more than 10,000 individuals (Supplementary Table S1). In these latter months, the samples were composed of 92,552 individuals of *T. pacificus* (73.0%) and 33,486 individuals of *P. azonus* (26.4%) in February, 71,176 individuals of *P. azonus* (69.2%) and 17,864 individuals of *T. pacificus* (17.4%) in March, and 89,616 individuals of *T. pacificus* (65.6%) and 28,468 individuals of *S. quinqueradiata* (20.8%) in October. In 2008, the number of individuals ranged from 390 in September to 14,078 in April (Figure 4). Further, January, September, and November 2008, recorded more than 10,000 individuals (Supplementary Table S1) In these latter months, the samples were composed of 95,440 individuals of *T. pacificus* (92.5%) and 4882 individuals of *S. quinqueradiata* (4.7%) in January, 108,431 individuals of *T. pacificus* (64.2%) and 53,184 individuals of *S. japonicus* (31.5%) in September, and 56,880 individuals of *T. japonicus* (40.0%) and 23,391 individuals of *S. quinqueradiata* (15.3%) in November (Supplementary Table S1).

The peaks in species diversity were recorded in June (H' = 1.59) and November (H' = 1.49) in 2007, and in April (H' = 1.72) and December (H' = 1.34) in 2008. The lowest diversity was recorded in February (H' = 0.21) in 2007 and January (H' = 0.41) in 2008. The diversity index gradually increased or repeatedly increased and decreased from March to June, then decreased from July to September, and subsequently, increased from October to November for both 2007 and 2008 (Figure 4).

*3.4. Analysis of the Fish Community*

The cluster analysis for the species composition and related time points showed three distinct time points in the Q-mode analysis and three groups of fish assemblages in the R-mode analysis.

The Q-mode analysis divided the time points into three significantly different groups (Figure 5): Group A comprised February, March, and April of both 2007 and 2008, along with May 2007. Group B comprised May 2008, and June and July of both 2007 and 2008. Group C comprised August and January 2008, and September, October, November, and December of both 2007 and 2008.

The R-mode analysis of fish assemblages formed three groups (Figure 5): Group A includes *C. saira*, *Acanthopagrus schlegeli*, and *Konosirus punctatus* that sporadically appeared from April to July, and from October to December for both 2007 and 2008; and *Hyperoglyphe japonica*, *Thamnaconus modestus*, and *T. japonicus* that appeared temporarily from August to

December. Group B included *Tribolodon taczanowskii*, *Lophiomus setigerus*, *Sebastes schlegelii*, *Takifugu chinensis*, *Paralichthys olivaceus*, *Mugil cephalus*, and *S. cirrhifer*, all of which appeared for a long duration during the entire study period, except from July to September. In addition, this group included *P. azonus* and *Oncorhynchus masou* that appeared from February to July. Group C included *S. quinqueradiata*, which intensively appeared from August to December, and *T. pacificus*, which appeared throughout the year.

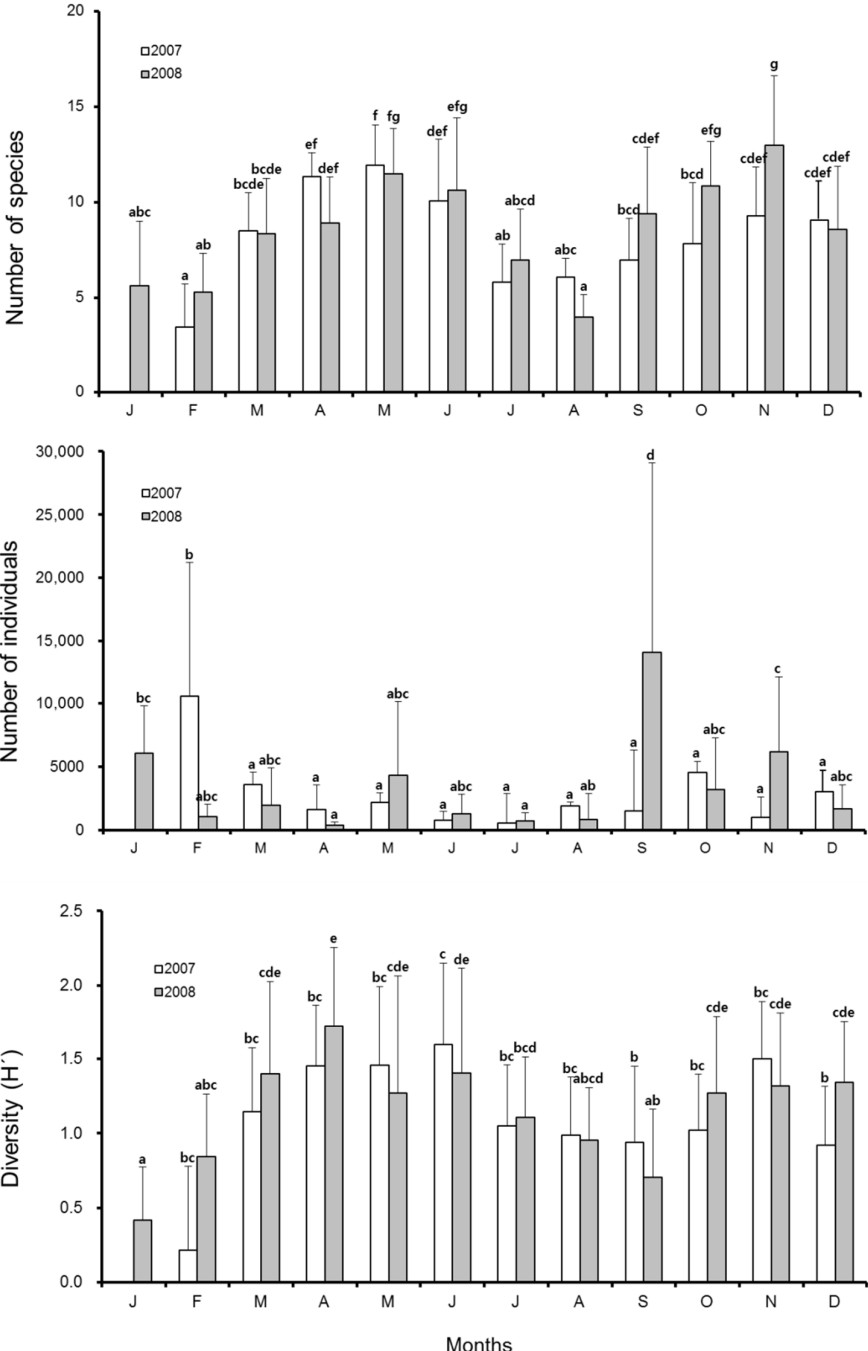

**Figure 4.** Monthly variations in the number of species, the number of individuals, and the diversity (H') of fishes captured in set nets in the Yangyang coastal areas from February 2007 to December 2008. The significance of the differences was assessed by one-way ANOVA followed by a Tukey's test. Different letters above each error bar indicate statistically significant differences compared to control conditions at $p < 0.05$.

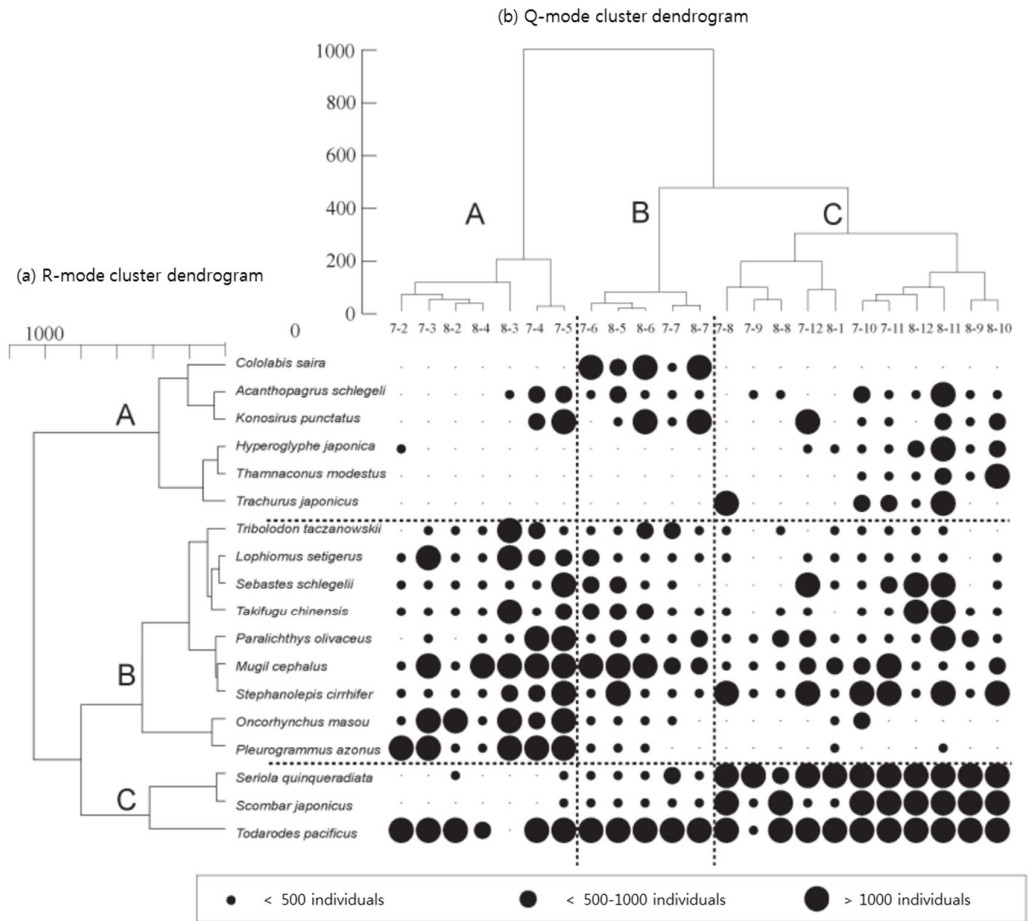

**Figure 5.** Cluster dendrograms based on the cluster analysis and relative abundance of the 18 most dominant fish species from 2007 to December 2008. (**a**) R-mode cluster dendrogram and (**b**) and Q-mode dendrogram. Monthly expressions are simplified in the form as 7–2 (February 2007).

## 4. Discussion

A total of 51 fish species were identified during the study period (Table 1). These species can be divided into 19 pelagic fish species, including *T. Pacificus*, *S. quinqueradiata*, and *S. japonicus* and 32 semi-demersal and demersal fish species, such as *S. schlegelii*, *P. olivaceus*, and *P. azonus* (Supplementary Table S1). The set net method is a passive fishing method that involves waiting for the fish to appear. It is strongly influenced by the local environmental conditions that directly affect the arrival of fish and their movements. This method can provide significant information on the distribution and characteristics of migratory fish, mainly because these species are highly mobile and travel in clusters [20,21]. This study provides data on the seasonal composition characteristics of migratory pelagic fish species, which have a high catch rate. The environmental factors affecting the distribution of fish, include water temperature, salinity, predators, and food biomass [22]. The water temperature and food biomass had been proposed as factors affecting the distribution of fish in the coastal waters of Korea [23].Salinity affects the density of horse mackerel in the South Sea of Korea, and this increase has been suggested as a factor [7,9]. Based on these studies, it is expected that the seasonal species composition data of this study can provide basic information for identifying the correlation between the fish species composition and the seasonal environmental factor changes on the Eastern coast of South Korea.

The monthly variations in the number of species peaked in May and November, when the water temperature increased and decreased, respectively (Figures 3 and 4). The number of species was higher in spring (April–June) and autumn (October–December)



than in summer (July–September) and winter (January–March) during the study period. These results were similar to those obtained in a previous study that used gill nets in the coastal waters off Shinsudo in Shamchonpo, on the south of the east coast of the Korean Peninsula [24]. Previous studies that conducted surveys in November, February, May, and August and involved the use of set nets reported that the number of species peaked in November in the coastal waters of Hupo, Gyeongsanbuk-do, south of the east coast of Korea, and in the coastal waters of Jangho, Gangwon-do [25]. In addition, Ryu et al. (2005) observed a peak in October in Oho of the Gangwon-do coastal waters during surveys conducted in May, August, October, and January. During the study period, the two peaks with the highest number of species in May and November, were evidently influenced by the entry of migratory fish species, such as *Engraulis japonicus*, *Konosirus punctaus*, *Seriola lalandi*, *Seriola quinqueradiata*, *Scomber japonicas*, and *Scomberomorus niphonius* (Figure 4 and Supplementary Table S1) [11].

Previously, 34–89 fish species were captured in a set net on the east coast of Korea in the Gyeongsangbuk-do coastal waters from 1993 to 2005 and 39–103 species were captured in the Gangwon-do coastal waters from 1998 to 2005 (Table 3). In this study, 49 and 46 species were recorded in 2007 and 2008, respectively, in Yangyang, indicating a low level of appearance compared with that observed in the previous studies. The number of species in each region varied significantly; however these variations could not be examined comprehensively based on the time and location of the observations (Table 3). Regarding the previous seasonal variations of the dominant species, these fish species mainly appeared on the Yangyang coast during winter. *C. pallasii* was the dominant species during 1998–2003, specifically from winter to spring in 2002. Further, *Lophius litulon* started appearing along with *C. pallasii*, and in 2003. *T. pacificus* was also one of the dominant species from 2005 onwards. This species was also recorded as a major dominant species in the present study (Table 3). In spring, *M. cephalus* was dominant from 2000 to 2002, *P. azonus* and *M. cephalus* were dominant in 2003 and 2005, and *T. pacificus* and *M. cephalus* were dominant in Yangyang in this study (Table 3). In the summer of 1998, *S. quinqueradiata* and *T. japonicus* alternately appeared as dominant species from 1998 to 2003; however, *T. pacificus* and *S. quinqueradiata* dominated in 2005. Similarly, these two species were also recorded as the dominant species in this study (Table 3). Following the autumn of 1998, *T. pacificus* was repeatedly recorded as the dominant species, and this trend continued throughout the study period (Table 3). Further, the catch period of *T. pacificus* was remarkably extended, making it a dominant species in all seasons after 2005. *T. pacificus* is a major commercial species widely distributed in the waters around Korea and Japan, and shows extensive seasonal migrations [26,27]. Moreover, the population of *T. pacificus* is known to increase when the spawning area increases with increasing water temperatures (18–23 °C) [28]. The water temperatures of Yangyang during the study period were higher by 0.1–1.8 °C in 2007 and 0.6–2.1 °C in 2008 than the average water temperature during the previous decade (Figure 3). Therefore, it is presumed that the high water temperature observed during this study increased the spawning area of *T. pacificus*, thereby resulting in this species being dominant in all seasons.

**Table 2.** Seasonal variations of the dominant species of fishes captured in set nets from the East Sea of Korea.

| Study Area | Year | No. of Species | Dominant Species | | | | Reference |
|---|---|---|---|---|---|---|---|
| | | | Winter (Jan.–Mar.) | Spring (Apr.–Jun) | Summer (Jul.–Sep.) | Autumn (Oct.–Dec.) | |
| Gangwon-do | | | | | | | |
| Goseong | 2005 | 73 | *Mugil cephalus, Hemitripterus villosus, Lophius litulon* | *Pleurogrammus azonus, Mugil cephalus* | *Todarodes pacificus* | *Psenopsis anomala, Pleurogrammus azonus, Arctoscopus japonicus* | [29] |
| Yangyang | 1998 | 103 | *Oncorhynchus gorbuscha, Clupea pallasii* | *Pleurogrammus azonus, Seriola quinqueradiata, Acanthopagrus schlegeli* | *Cleisthenes pinetorum, Seriola quinqueradiata* | *Stephanolepis cirrhifer, Sebastes schlegelii, Todarodes pacificus* | [4] |
| | 1999 | 64 | *Clupea pallasii, Thamnaconus modestus* | *Todarodes pacificus, Thamnaconus modestus* | *Thamnaconus modestus* | *Konosirus punctatus, Stephanolepis cirrhifer, Trachurus japonicus* | [4] |
| | 2000 | 56 | *Clupea pallasii* | *Mugil cephalus, Clupea pallasii, Acanthopagrus schlegeli* | *Mugil cephalus, Trachurus japonicus* | *Stephanolepis cirrhifer, Todarodes pacificus, Ditrema temminckii* | [4] |
| | 2001 | 40 | *Clupea pallasii, Pleurogrammus azonus, Mugil cephalus* | *Mugil cephalus, Stephanolepis cirrhifer, Seriola quinqueradiata* | *Seriola quinqueradiata* | *Seriola quinqueradiata, Oncorhynchus masou, Todarodes pacificus* | [4] |
| | 2002 | 54 | *Lophius litulon, Clupea pallasii, Aptocyclus ventricosus* | *Mugil cephalus, Lophius litulon, Cleisthenes pinetorum* | *Trachurus japonicus, Scomber japonicus* | *Acanthopagrus schlegeli, Konosirus punctatus* | [4] |
| | 2003 | 39 | *Lophius litulon, Clupea pallasii* | *Pleurogrammus azonus, Mugil cephalus, Lophius litulon* | *Trachurus japonicus, Thamnaconus modestus* | *Psenopsis anomala, Trachurus japonicus* | [4] |
| | 2005 | 70 | *Todarodes pacificus, Clupea pallasii, Lophius litulon* | *Pleurogrammus azonus, Todarodes pacificus, Mugil cephalus* | *Todarodes pacificus, Seriola quinqueradiata* | *Scomber japonicus, Engraulis japonicus, Mugil cephalus* | [29] |
| | 2007 | 49 | *Todarodes pacificus, Pleurogrammus azonus, Mugil cephalus* | *Todarodes pacificus, Pleurogrammus azonus, Mugil cephalus,* | *Scomber japonicus,Todarodes pacificus, Stephanolepis cirrhifer* | *Todarodes pacificus, Seriola quinqueradiata, Stephanolepis cirrhifer* | Present study |

**Table 3.** Seasonal variations of the dominant species of fishes captured in set nets from the East Sea of Korea.

| Study Area | Year | No. of Species | Dominant Species | | | | Reference |
| --- | --- | --- | --- | --- | --- | --- | --- |
| | | | Winter (Jan.–Mar.) | Spring (Apr.–Jun) | Summer (Jul.–Sep.) | Autumn (Oct.–Dec.) | |
| | 2008 | 46 | *Todarodes pacificus, Pleurogrammus azonus, Oncorhynchus masou* | *Todarodes pacificus, Mugil cephalus, Stephanolepis cirrhifer* | *Todarodes pacificus, Scomber japonicus, Seriola quinqueradiata* | *Todarodes pacificus, Trachurus japonicus, Seriola quinqueradiata* | Present study |
| Gyeongsangbuk-do | | | | | | | |
| Uljin | 2005 | 50 | *Todarodes pacificus, Konosirus punctatus, Tribolodon taczanowskii* | *Todarodes pacificus* | *Todarodes pacificus, Scomber japonicus* | *Auxis rochei, Konosirus punctatus, Todarodes pacificus* | [29] |
| Heunghae | 1993 | 61 | *Todarodes pacificus, Etrumeus teres* | *Todarodes pacificus, Engraulis japonicus* | *Todarodes pacificus, Tamnaconus modestus, Trachurus iaponicus* | *Scomber japonicus, Loligo bleekeri, Todarodes pacificus* | [30] |
| Heunghae | 1994 | 57 | *Todarodes pacificus, Pampus echinogaster, Loligo bleekeri* | *Todarodes pacificus, Cololabis saira* | *Scomber japonicus, Trachurus japonicus* | *Todarodes pacificus, Trachurus japonicus, Trichiurus lepturus Trachurus iaponicus Trichiurus lepturus* | [30] |
| Heunghae | 1995 | 86 | *Todarodes pacificus, Clupea pallasii* | *Todarodes pacificus, Engraulis japonicus* | *Trachurus iaponicus, Scomber japonicus* | *Todarodes pacificus, Trachurus iaponicus, Scomber japonicus* | [31] |
| Guryongpo | 2001 | 34 | *Konosirus punctatus, Ditrema temminckii, Mugil cephalus* | *Ditrema temminckii, Mugil cephalus, Todarodes pacificus* | *Todarodes pacificus, Seriola quinqueradiata, Mugil cephalus* | *Todarodes pacificus, Mugil cephalus, Seriola quinqueradiata* | [4] |
| Guryongpo | 2003 | 79 | – | *Engraulis japonicus, Trachurus japonicus* | *Trachurus japonicus, Engraulis japonicus* | *Engraulis japonicus, Trachurus japonicus* | [4] |
| Guryongpo | 2005 | 53 | *Engraulis japonicus, Mugil cephalus* | *Engraulis japonicus, Trachurus japonicus* | *Trachurus japonicus, Engraulis japonicus* | *Engraulis japonicus, Sebastes schlegelii, Todarodes pacificus* | [29] |
| Ulsan | 1998 | 89 | *Engraulis japonicus, Pampus echinogaster* | *Cololabis saira, Pleurogrammus azonus, Engraulis japonicus* | *Scomber japonicus, Trachurus japonicus* | *Trachurus japonicus, Tamnaconus modestus, Scomber japonicus* | [32] |

(–) denotes the lack of data. Figure legends.

The monthly variations in the number of individuals in each species were high in winter and autumn. The highest daily average number of appearances was observed in February 2007 and September 2008. In addition, the cumulative number of individuals exceeded 10,000 in February, March, and October 2007, and in January, September, and November 2008. During this period, the cumulative population of the first and second dominant species was 55.3% to 99.4%, respectively. *T. pacificus* was the dominant species during the entire study period, followed by *P. azonus* in February and March 2007, *S. quinqueradiata* in October 2007 and January 2008, and *S. japonicus* and *S. quinqueradiata* in September and November 2008. *P. azonus* has a high catch record, and is a dominant species from winter to spring on the east coast of Korea [4,29,33]; however, this species was the second dominant species that appeared from February to April 2007, and March 2008 (Supplementary Table S1). These observations were similar to those reported in previous studies [4,29]. In addition, *P. azonus* influenced the number of individual fishes, such as *T. pacific*, *S. japonicus*, and *S. quinqueradiata*, that were classified as small and large pelagic fish in Korean waters [8]. These species, along with *P. azonus*, are affected by warm waters in winter. Further, the trends in the changes in the diversity index were similar to those of the monthly variations in the number of species. A high number of species was observed in spring and autumn, with equal proportions of many species. Moreover, a low number was observed in winter and summer, with some species observed in large proportions.

The dominant species (more than 0.4% of the total number of observed species) were divided into three groups through a cluster analysis. Group A included species that appeared temporarily and sporadically in spring to autumn, and those that appeared intensively in summer to autumn (Figure 5). In this group, *C. saira* and *T. japonicus* were representative of the migratory small pelagic fishes of the East Sea that migrate north in spring and summer and south in autumn and winter [4,34]. *A. schlegeli* and *K. punctatus* were observed in spring and autumn and were found to contribute to increasing the size of the community, similarly with *C. saira*, that had a high number of catches in spring, and with other main catch species, such as *H. japonica*, *T. modestus*, and *T. japonicas*, in autumn. Group B comprised fish species that appeared for a long duration and almost the entire study period, except in summer (July to September); additionally, this group included species that appeared intensively from winter to spring. In this group, the small pelagic fish species, such as *M. cephalus* and *S. cirrhifer*, whose distributions are highly affected by warm currents [4,34] were found throughout the study period. Further, resident species, such *as T. taczanowskii*, *L. setigerus*, *S. schlegelii*, *T. chinensis*, *P. olivaceus*, *M. cephalus*, and *S. cirrhifer*, were observed during the study period and were placed in the same group as *O. masou* and *P. azonus*, which appeared intensively from winter to spring. Group C consisted of species that accounted for 64% of the total species (Supplementary Table S1), and were recorded throughout the study period or intensively from summer to autumn. This group comprised species, such as *T. pacificus*, *S. quinqueradiata*, and *S. japonicus*, which are representative of warm current migratory fish species travelling towards the south/north depending on the season; additionally, these species typically migrate seasonally along the east coast of the Korean Peninsula [32,35–38]. Such an extensive distribution is largely dependent on the annual changes in the sea conditions [39]. *T. pacificus* was a migratory species that consistently appeared throughout the entire study period, and also represented the major residing species. *S. quinqueradiata* and *S. japonicus* appeared throughout the entire period, except from February to April (Figure 5). An increase or decrease in fish resources is related to the water temperature and periods of plankton growth [40]; additionally, it can be presumed that there is a difference in the distribution of migratory species depending on the influence of a warm current, which in turn is a critical aspect that should be considered during the study period [32]. In previous studies, the water temperature was observed to increase by 2.59 °C in the surface waters of Tsushima and by 0.35 °C in the East Sea waters [14]. In addition, the water temperature at a depth of 20 m, as observed through the satellite image data (NOAA/AVHRR), increased by 0.83 °C in the East Sea from 2000 to

2009 [41]. Therefore, the increased water temperature during this study as confirmed by our observations would have affected the fish community structure.

The analysis of the seasonal variations in the species composition of fishes captured along the Yangyang coast in the central part of the East Sea of Korea indicated that major migratory pelagic fishes were the dominant species throughout the study period and showed high catch rates. It is presumed that the period when the water temperature in this area is strongly influenced by the warm current, along with other factors, can affect the appearance of pelagic fishes; however, understanding the overall characteristics of fish resources using only selective data regarding the pelagic fish species from the catchment nets and the analysis of the integrated logbooks data of three set net points, that cannot be separated by each point, may act as a limitation. Nevertheless, the results of this study on the seasonal variations of the species compositions of set net fisheries can serve as important baseline information for improving the predictability of future changes in the fishery resources, especially concerning the type of fish caught according to the seasonal changes in the water temperature in the region.

In order to provide more accurate information about the seasonal variations in the composition of the pelagic and demersal fish species, it is necessary that an assessment be conducted on the types of fishing gear that can monitor the lower layers of the water column and to analyze this data together with the changes in the environmental factors such as salinity and food abundance.

**Supplementary Materials:** The following supporting information can be downloaded at: https://www.mdpi.com/article/10.3390/jmse10081102/s1, Table S1: The numbers of individuals of fishes caught by the set net in the coastal waters of Yangyang from February 2007 to December 2008.

**Author Contributions:** Data curation, formal analysis, writing—original draft, J.H.; Conceptualization, interpretation of results and discussion, writing—original draft, Y.-U.C. All authors have read and agreed to the published version of the manuscript.

**Funding:** This work was supported by the Korea Institute of Energy Technology Evaluation and Planning (KETEP), the Ministry of Trade, Industry & Energy (MOTIE) of the Republic of Korea (No. 20203040020130).

**Institutional Review Board Statement:** Not applicable.

**Informed Consent Statement:** Not applicable.

**Data Availability Statement:** All data generated or analyzed during this study are available via the data repository of the KIOST. Requests for material should be made to the corresponding author.

**Acknowledgments:** The authors would like to thank to support for the Korea Institute of Energy Technology Evaluation and Planning (KETEP), the Ministry of Trade, Industry & Energy (MOTIE) of the Republic of Korea (No. 20203040020130). Finally, we thank the editor and the anonymous reviewers whose comments greatly improved the manuscript.

**Conflicts of Interest:** The authors declare no conflict of interest.

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
