# Peer review of "Seasonal Variations in the Fish Species Composition and Community Structures on the Eastern Coast of South Korea"

_jmse, doi:10.3390/jmse10081102_

Round 1

Reviewer 1 Report

1.  The manuscript contains the capture data of set nets fisheries from 2007 to 2008, and discusses with relevant literature. At the same time, it also analyzes the correlation changes between seawater temperature and fish species. The results are of academic value, but the data period is relatively earlier.

2.  Figure 4 shows that the number of species is highly correlated with diversity, but most months are negatively correlated with the number of individuals. What may be the reason for this? In addition, the number of individuals in February 2007 and September 2008 was very high, and The gap with other months is obvious (volatility). Are there other factors affecting the presentation of this part of the data?

3. Table 2 fully discusses the dominant species in the past literature in different seasons, but are there different ways of sampling or data collection in different studies?

4. During the study period, t. pacificus became the dominant species, and the capture of p. azonus was very high. The results of the study also found that the temperature was related to the seasonal variation of the capture. However, due to the earlier data period (2007-2008), considering the effect of climate change in recent years. It is believed that it would be more meaningful if the recent captures could be disclosed and discussed.

Author Response

  1. The manuscript contains the capture data of set nets fisheries from 2007 to 2008, and discusses with relevant literature. At the same time, it also analyzes the correlation changes between seawater temperature and fish species. The results are of academic value, but the data period is relatively earlier.

Response: Thank you for the comments. We absolutely agree with your comment. Unfortunately, there is not enough information on fishing data from the late 2000s on the eastern coast of Korea, so it became very important to provide information during this period. The information provided here is not entirely new but fills gaps in the current knowledge. Moreover, retrospective analysis has been mentioned in the discussion. The significance of the findings in this study is the catch data of the late 2000s, and it is judged to be data that connects variations of the past, present, and future in the appearance of major fish species, and it helps to improve the predictive patterns in the future. Therefore, despite the passage of time, we would like to provide information by submitting the data to the scientific journal.

  1. Figure 4 shows that the number of species is highly correlated with diversity, but most months are negatively correlated with the number of individuals. What may be the reason for this? In addition, the number of individuals in February 2007 and September 2008 was very high, and The gap with other months is obvious (volatility). Are there other factors affecting the presentation of this part of the data?

Response: Thank you for the comments. The previous reports were indicated that the diversity index statistical representations of biodiversity in different aspects such as species richness and evenness. In February 2007 and September 2008, the number of individuals was large, but the number of species was small compared to other months, which could be the main factor contributing to the low species richness, species evenness, and diversity. In supplementary data, species richness and species evenness were lower than in other months and the diversity index was also low, as follows; 1.40 of species richness, 0.081 of species evenness, 0.215 of diversity index in February 2007 and 2.19 of species richness 0.227 of species evenness, 0.703 of diversity index in September 2008. (Note: Although the number of individuals that appeared in March 2007 was small, but species richness was 2.56, species evenness was 0.433, and species diversity was 1.143).

  1. Table 2 fully discusses the dominant species in the past literature in different seasons, but are there different ways of sampling or data collection in different studies?

Response: The results from previous studies of the data in Table 2 are the fishing results of the same set net as the fishing gear used in this study. In discussion of Table 2, the change patterns with the species composition and number of individuals were discussed by fishing time and fishing location on eastern costal of Korea.

  1. During the study period, t. pacificus became the dominant species, and the capture of p. azonus was very high. The results of the study also found that the temperature was related to the seasonal variation of the capture. However, due to the earlier data period (2007-2008), considering the effect of climate change in recent years. It is believed that it would be more meaningful if the recent captures could be disclosed and discussed.

Response: Thank you for your comments. We believe these are important points and we concur with your suggestions. We believe that your valuable suggestion can be included in our future study. Since 2007, we have been using this thesis as a starting point and we are in the process of constructing data and writing manuscripts to analyze the correlation between water temperature, species composition, and diversity index. In the near future, as you have suggested, we will be able to propose a manuscript with more abundant data and more detailed information. Thanks again for your great suggestion.

Reviewer 2 Report

My only suggestion is to revise the English of the present manuscript by a native.

Author Response

My only suggestion is to revise the English of the present manuscript by a native.

Response: Thank you for your suggestion. The grammar was submitted for revision to a company specializing in English manuscript proofreading.

Reviewer 3 Report

The manuscript contributes recent information about the composition of the fish fauna in front of a given part of the Korean marine sector. The information is not entirely new but fills gaps in the current knowledge. Moreover, retrospective analysis has been added to the discussion. As a whole, it is well written and presented, few issues have to be improved, e.g. last statements as conclusions. A single question arose: why the authors forwarded the manuscript after 14 years? 

Author Response

The manuscript contributes recent information about the composition of the fish fauna in front of a given part of the Korean marine sector. The information is not entirely new but fills gaps in the current knowledge. Moreover, retrospective analysis has been added to the discussion. As a whole, it is well written and presented, few issues have to be improved, e.g. last statements as conclusions. A single question arose: why the authors forwarded the manuscript after 14 years?

Response: Thank you for the comments. We absolutely agree with your comment. Unfortunately, there is not enough information on fishing data from the late 2000s on the eastern coast of Korea, so it became very important to provide information during this period. The information is not entirely new but fills gaps in the current knowledge. Moreover, retrospective analysis has been mentioned to the discussion. The findings of this study is the catch data of the late 2000s, and it is judged to be data that connects the past, present, and future of variability in the appearance of major fish species, and it can help improve the predictive patterns of the future. Therefore, despite the passage of time, we would like to provide information by submitting the data to the scientific journal.

Reviewer 4 Report

The manuscript fits within the scope of the Journal of Marine Science and Engineering but some major changes need to be done. Namely, as I can understand authors investigated species composition and community structure in the area of coastal waters of Yangyang, Gangwon-do coast, caught with set nets from February 2007 to December 2008. Authors compare obtained results between these two years even though they stated that sea temperatures measured in this investigated period were the highest in the last decade. So, I would expect that they will statistically compare the composition and structure of the community with the ones obtained before 2007 since according to the reference list they have the data (calculate H' index, Evenness (E), Jaccard Index during the previous years and compare them). Furthermore, I would like to see some correlation between sea temperature and species composition or structure (for example correlation between sea temperature and H' index). I think that the authors put a lot of effort to obtain this data set but they can provide us with better and much more interesting results. Since I am not a native speaker I will recommend the authors check the English language.

Author Response

The manuscript fits within the scope of the Journal of Marine Science and Engineering but some major changes need to be done. Namely, as I can understand authors investigated species composition and community structure in the area of coastal waters of Yangyang, Gangwon-do coast, caught with set nets from February 2007 to December 2008. Authors compare obtained results between these two years even though they stated that sea temperatures measured in this investigated period were the highest in the last decade. So, I would expect that they will statistically compare the composition and structure of the community with the ones obtained before 2007 since according to the reference list they have the data (calculate H' index, Evenness (E), Jaccard Index during the previous years and compare them). Furthermore, I would like to see some correlation between sea temperature and species composition or structure (for example correlation between sea temperature and H' index). I think that the authors put a lot of effort to obtain this data set but they can provide us with better and much more interesting results. Since I am not a native speaker I will recommend the authors check the English language.

Response: Thank you for your great suggestion. In this study, the range values represent the minimum and maximum values of the monthly water temperature in the past 10 years and the monthly water temperature difference in 2007 and 2008, during the study period. The focus was to confirm effects of seasonal variations on fish species composition and community structure in 2007 and 2008, during the study period. Unfortunately, as you have suggested, we planned to compare species composition and community structure before 2007, but the previous data in previous studies were not sufficient, and there was a limitation in comparison in this paper because only information limited to the number of species and dominant species in the form of a report was provided. Currently, since 2007, using this thesis as a starting point, we are in the process of constructing data and writing manuscripts to analyze the correlation between water temperature and species composition, diversity index. We hope that in the near future, like your suggestion, we will be able to propose a manuscript with more abundant data and more detailed information. Thanks again for your great suggestion.

Round 2

Reviewer 1 Report

The author has a complete reply, which can fully understand the difficulty of collecting fishery data. After all, the research period of the data is relatively long. Therefore, if author can discuss with some recent data, it is believed that the reference value can be improved.

This manuscript is a resubmission of an earlier submission. The following is a list of the peer review reports and author responses from that submission.

Round 1

Reviewer 1 Report

This manuscript presents the seasonal variability of fish in the coastal waters of South Korea. The authors analyse changes in fish species composition, abundance and biodiversity.

The manuscript is relatively simple. The strengths of this manuscript are the numerous research materials and interesting cluster analyses. The weaker points of this manuscript are the inference limited to local waters. The majority of the references are local papers published in Korean journals and reports. I see the need to include local data and spread it to international readers. Especially since many of them are only available to those who know Korean. However, using only such references in the manuscript is a mistake.

Cluster analysis is the only method used by the authors. I would like to see the results of statistical analyses comparing the monthly variation of parameters describing fish in both years.

The discussion has the character of a description of the results, at most with a confrontation of other studies. I encourage the authors to speculate more on the causes of the observed fish variability in the waters of the eastern coast of South Korea.

Specific comments:

Line 13: please check the water temperature information. Figure 1 shows other data.

Line 24: some keywords are repeated with the title. The authors should correct these words.

Lines 59-61: Authors should provide more details, such as the depth of the fishing site.

Line 66: What does species levels mean?

Lines 70-71: At what depth was the water temperature measured?

Lines 79-82: Please give a little more detail on how these analyses were done.

Line 150: Why is there no variability in December 2007?

Line 241: What effect did the different fishing gears have on the data shown in the table?

Author Response

Response to Reviewers’ Comments

We greatly appreciate the valuable inputs of the anonymous reviewers on our work and thank them for their valuable suggestions on the manuscript. According to their comments, we have thoroughly revised the manuscript and incorporated changes as suggested by the reviewers. The revisions are marked in red in the revised manuscript.

Reviewer #1

This manuscript presents the seasonal variability of fish in the coastal waters of South Korea. The authors analyses changes in fish species composition, abundance and biodiversity. The manuscript is relatively simple. The strengths of this manuscript are the numerous research materials and interesting cluster analyses. The weaker points of this manuscript are the inference limited to local waters. The majority of the references are local papers published in Korean journals and reports. I see the need to include local data and spread it to international readers. Especially since many of them are only available to those who know Korean. However, using only such references in the manuscript is a mistake. Cluster analysis is the only method used by the authors. I would like to see the results of statistical analyses comparing the monthly variation of parameters describing fish in both years. The discussion has the character of a description of the results, at most with a confrontation of other studies. I encourage the authors to speculate more on the causes of the observed fish variability in the waters of the eastern coast of South Korea.

Response: Thank you for your comments. We agree with your suggestion that the information on statistical analyses, comparing the monthly variation of parameters describing fish in both years, is valuable. We have made corrections in the revised Figure 4 to highlight this. In addition, as suggested, we have modified several sentences in the text to highlight this.

Specific comments:

Review1-1) Line 13: please check the water temperature information. Figure 1 shows other data.

Response: Thank you for your suggestion. We have modified the sentence in the revised manuscript (“The water temperature of the study area was 0.1–2.1 °C during the study period, which was higher than the average water temperature in the last decade.” “The water temperature of the study area was 0.1–1.8 °C in 2007 and 0.6–2.1 °C in 2008, which was higher than the average water temperature in the last decade”). In addition, we clarify that Figure 1 is a map showing the location of the set nets in the coastal waters of Yangyang. Figure 3 shows the monthly variations in the mean water temperature during the study period (2007–2008) in the coastal waters of Yangyang, and in the last decade (1999–2008) in the coastal waters of Sokcho.

Review1-2) Line 24: some keywords are repeated with the title. The authors should correct these words.

Response: Thank you for your suggestion. We have corrected this in the revised manuscript.

Review1-3) Lines 59-61: Authors should provide more details, such as the depth of the fishing site.

Response: Thank you for your suggestion. We have corrected this in the revised manuscript (The point of installation of set nets was approximately 3.8 km from the coastline, and the water depth was approximately 50 m).

Review1-4) Line 66: What does species levels mean?

Response: We apologize for the confusion. We have clarified this in the revised manuscript (species levels-> species diversity).

Review1-5) Lines 70-71: At what depth was the water temperature measured?

Response: The water temperature data in manuscript is sea surface temperature data. We have specified this in the revised manuscript (water temperature data -> sea surface temperature).

Review1-6) Lines 79-82: Please give a little more detail on how these analyses were done.

Response: Thank you for your suggestion. We have added these details in the revised manuscript

Review1-7) Line 150: Why is there no variability in December 2007?

Response: We apologize for the confusion. We have revised Figure 4.

Review1-8) Line 241: What effect did the different fishing gears have on the data shown in the table?

Response: Figure 2 shows information regarding fishing gear (we used only one fishing gear) and further details are mentioned in the discussion section (The set net method is a passive fishing method that involves waiting for the fish to appear. It is strongly influenced by local environmental conditions that directly affect fish arrival and their movements. This method can provide significant information on the distribution and characteristics of migratory fish, mainly because these species are highly mobile and travel in clusters). In this regard, our study showed the results of the number of species, number of individuals, and diversity.

Reviewer 2 Report

The authors provided an interesting observation on the variability in fish community in an area greatly exploited for commercial fishery purposes. However, the analyses carried out as well as the discussions about are lacking and must be improved. For instance, at the end of the discussions the authors suggested the usefulness of these results for a sustainable management of fishery. In my opinion, they should clarify if this seasonally variability in presence of fish species is associated to a variability in fishery activities (i.e. differences in fishery gears, vessels sizes, etc.) or if it may be an objective to gain in the study area. It means, it is not clear for the reader if the management plans should be improved or totally introduced in the study area. The knowledge of this information may determine a different way to speculate on the obtained results, in my opinion.

The seasonal variability or, at least, a variability in fish community among months or associated with seasonal changeability in temperature and environmental conditions have been already observed, as the authors showed in the discussions. Such information should be already provided within the introduction and later discussed highlighting differences or improving of knowledge respect to the previous observations.

Why the variability among the three sampling points have not been analyzed? Are they too close among them, thus showing the same environmental conditions? Or are not data enough for a robust analysis? Furthermore, even different depths or characteristics of the bottom among the sampling sites could lead to differences in the community composition. The differences among the sampling sites should be evaluate or the reasons making this analyses impossible or not consistent explained.

Finally, the English must be revised by a native speaker. Some sentences must be rewritten in order to avoid confusion and be more easily readable.

Please, find below some minor comment.

L. 35-40 please, rephrase. The sentence appeared very confused. Moreover, the numerous repetitions of “fishery production” contribute in making difficultly readable such sentence.

L. 47 The authors should briefly explain what “Kuroshio” is and its influence on oceanography and ecology of the study area.

L 50-51 Please, add references about such evidences.

L 59-62 information about the distance from the coast and depth at which set nets were operating would be useful to provide complete knowledge on sampling.

L89-90 Delete this sentence. The reference to the Figure 3 at the end of the paragraph is enough.

L.91 range of temperature or mean values per year? Please, specify.

L 92-92. Why the authors referred to March –May and June –August separately, they are contiguous periods. Would the authors just highlight the data were collected seasonally or there is some difference in mean water temperature between these quarters? The sentence seems to reported similar water temperature in both quarters (seasons). Please, rephrase to better clarify this argument.

L 96-98 What the range values (i.e. 0.1–1.8 °C and 0.6–2.1 °C) mean? The authors calculated a monthly variability and the ranges showed minimum and maximum of variability per year? Please, explain.

L 122 not clear. Please, rephrase.

L 130-134 Please, modified as follow: “In these latter months, the samples were composed by 92,552 individuals of T. pacificus (73.0%) and 33 486 131 individuals of P. azonus (26.4%) in February, 71 176 individuals of P. azonus (69.2%) and 132 17 864 individuals of T. pacificus (17.4%) in March….”. Similarly, for L 137-140.

L 177-185 Move to the Introduction or, at least, synthetize and put later on L 190-192, when the authors referred to the strongly influence of environmental conditions.

L 187-189 Why the authors mentioned particularly these species? They are not the fish species cited into the Introduction as the most commercially exploited. Probably, they have been chosen for other reasons, e.g. ecologically importance, wider distribution in the study area… Please, specify and produce proper reference if available.

L 196 and subsequent. Authors may provide some explanation, as suggestion/hypothesis as well, about such seasonality in peaking of number of specie, i.e. temperature/season affects the enrichment in nutrients, the availability of prey, etc. Several studied in literature can provide good suggestions about.

L 270 and L 285 “representative of..”

Table 1 The layout of the table should be modified. “Cephalopods” would be better in bold type as “Fishes”, and maybe a black line separating fishes and cephalopods needs to be added. Moreover, the caption must to be revised: “Fish and cephalopod species captured…”

Table 2 need to revise the layout. For instance, to modify something like YEA-R, 200-5, etc.

Author Response

Response to Reviewers’ Comments

We greatly appreciate the valuable inputs of the anonymous reviewers on our work and thank them for their valuable suggestions on the manuscript. According to their comments, we have thoroughly revised the manuscript and incorporated changes as suggested by the reviewers. The revisions are marked in red in the revised manuscript.

Reviewer #2: The authors provided an interesting observation on the variability in fish community in an area greatly exploited for commercial fishery purposes. However, the analyses carried out as well as the discussions about are lacking and must be improved. For instance, at the end of the discussions the authors suggested the usefulness of these results for a sustainable management of fishery. In my opinion, they should clarify if this seasonally variability in presence of fish species is associated to variability in fishery activities (i.e. differences in fishery gears, vessels sizes, etc.) or if it may be an objective to gain in the study area. It means, it is not clear for the reader if the management plans should be improved or totally introduced in the study area. The knowledge of this information may determine a different way to speculate on the obtained results, in my opinion. The seasonal variability or, at least, a variability in fish community among months or associated with seasonal changeability in temperature and environmental conditions have been already observed, as the authors showed in the discussions. Such information should be already provided within the introduction and later discussed highlighting differences or improving of knowledge respect to the previous observations. Why the variability among the three sampling points have not been analyzed? Are they too close among them, thus showing the same environmental conditions? Or are not data enough for a robust analysis? Furthermore, even different depths or characteristics of the bottom among the sampling sites could lead to differences in the community composition. The differences among the sampling sites should be evaluated or the reasons making these analyses impossible or not consistent explained. Finally, the English must be revised by a native speaker. Some sentences must be rewritten in order to avoid confusion and be more easily readable.

Response: Thank you for your comments. We believe these are important points and we concur with your suggestions. The study on fish composition was conducted in same consideration of regional characteristics, and it can be said that the composition of each element presented the expected results in this study area, although direct comparison is difficult. The primary focus of this study was to confirm the effects of seasonal variations on fish species composition and community structure. The findings of this study can contribute to improving the future predictive power of species compositional characteristics due to seasonal changes in this region, especially changes in water temperature.Based on your insightful suggestion, this content has been marked in the manuscript to further clarify the expected effect of the results of this study.In addition, we have described the difficulty in analysis of the three set net fishing points individually at the end of the discussion.The answers to other minor comments are as follows:

Minor comments:

Review2-1) L. 35-40 please, rephrases. The sentence appeared very confused. Moreover, the numerous repetitions of “fishery production” contribute in making difficultly readable such sentence.

Response: Thank you for your suggestion. We have corrected this in the revised manuscript.

Review2-2) L. 47 The authors should briefly explain what “Kuroshio” is and its influence on oceanography and ecology of the study area.

Response: Thank you for your suggestion. We have added information on “Kuroshio” in the revised manuscript.

Review2-3) L 50-51 Please, add references about such evidences.

Response: Thank you for your suggestion. We have supplemented information on examples of warm water fish species and cited relevant references in the revised manuscript.

Review2-4) L 59-62 information about the distance from the coast and depth at which set nets were operating would be useful to provide complete knowledge on sampling.

Response: Thank you for your suggestion. We have included information on set net sampling points and depth of operation in the manuscript.

Review2-5) L89-90 Delete this sentence. The reference to the Figure 3 at the end of the paragraph is enough.

Response: Thank you for your suggestion. We have removed this sentence from the revised manuscript.

Review2-6) L.91 range of temperature or mean values per year? Please, specify.

Response: Thank you for your comment. The graph in Figure 3 shows the monthly average water temperature data. As suggested, it is indicated in the manuscript as monthly water temperature.

Review2-7) L 92-92 Why the authors referred to March –May and June –August separately, they are contiguous periods. Would the authors just highlight the data were collected seasonally or there is some difference in mean water temperature between these quarters? The sentence seems to reported similar water temperature in both quarters (seasons). Please, rephrase to better clarify this argument.

Response: Thank you for your comment. Initially, according to the curve of the graph, the intention was to express a gradual increase in March-May and a sharp increase in June-August. However, since this period is a continuous period, it is not necessary to divide it into two sections. We have revised March-May and June-August to March-August.

Review2-8) L 96-98 What the range values (i.e. 0.1–1.8 °C and 0.6–2.1 °C) mean? The authors calculated a monthly variability and the ranges showed minimum and maximum of variability per year? Please, explain.

Response: Thank you for your comment. The range values represent the minimum and maximum values of the monthly water temperature for the past 10 years and the monthly water temperature difference in 2007 and 2008, during the study period.

Review2-9) L 122 not clear. Please, rephrase.

Response: We apologize for the confusion. We have removed this in the revised manuscript.

Review2-10) L 130-134 Please, modified as follow: “In these latter months, the samples were composed by 92,552 individuals of T. pacificus (73.0%) and 33,486 individuals of P. azonus (26.4%) in February, 71,176 individuals of P. azonus (69.2%) and 17,864 individuals of T. pacificus (17.4%) in March….”. Similarly, for L 137-140.

Response: Thank you for your suggestion. We have corrected this in the revised manuscript.

Review2-11) L 177-185 Move to the Introduction or, at least, synthetize and put later on L 190-192, when the authors referred to the strongly influence of environmental conditions.

Response: Thank you for your suggestion. We have moved this to the last paragraph of the introduction. 

Review2-12) L 187-189 Why the authors mentioned particularly these species? They are not the fish species cited into the Introduction as the most commercially exploited. Probably, they have been chosen for other reasons, e.g. ecologically importance, wider distribution in the study area… Please, specify and produce proper reference if available.

Response: Thank you for your comment. The reason for mentioning these species was to refer to the representative pelagic fish and semi-demersal and demersal fish caught in the set nets in this study. Therefore, they were displayed in order of dominance. However, the first dominant species, T. pacificus, was missing in the pelagic fish; therefore, we have replaced T. japonicus with T. pacificus according to dominance. Pleuronectes schrenki was not included in the main fish species (Figure 5) in this study; therefore, it was changed to Pleurogrammus azonus according to dominance (Supplementary Table S1, Figure 5).

Review2-13) L 196 and subsequent. Authors may provide some explanation, as suggestion/hypothesis as well, about such seasonality in peaking of number of specie, i.e. temperature/season affects the enrichment in nutrients, the availability of prey, etc. Several studied in literature can provide good suggestions about.

Response: Thank you for your suggestion. We have supplemented this information in the revised manuscript.

Review2-14) L 270 and L 285 “representative of..”

Response: Thank you for your suggestion. We have added this in the revised manuscript.

Review2-15) Table 1 The layout of the table should be modified. “Cephalopods” would be better in bold type as “Fishes”, and maybe a black line separating fishes and cephalopods needs to be added. Moreover, the caption must to be revised: “Fish and cephalopod species captured…”

Response: Thank you for your suggestion. We have corrected this in the revised Table 1.

Review2-16) Table 2 need to revise the layout. For instance, to modify something like YEA-R, 200-5, etc.

Response: Thank you for your comment. We have revised Table 2. 

Round 2

Reviewer 2 Report

The authors provides an attentive revisions of the manuscript, as well as accurate replies to all the comments and suggestions. In my opinion, just to make the manuscript as most accurate as possible, the authors should provide in the Figure 4 a legend of letters within the graphs.